# What factors determine users' knowledge payment decisions? A mixed-method study

Qin Ying[1], Md. Mukitul Hoque[2], Sang-Joon Lee[2]*

**1** Henan Key Laboratory for Big Data Processing & Analytics of Electronic Commerce, Luoyang Normal University, Luoyang, China, **2** Interdisciplinary Program of Digital Future Convergence Service, Chonnam National University, Gwangju, South Korea

* s-lee@jnu.ac.kr

**Data Availability Statement:** The full data file associated with our study can be accessed at the following figshare repository: [Insert DOI: https://doi.org/10.6084/m9.figshare.23507724.v2].

## Abstract

Methods for obtaining valuable knowledge from the vast amount of mixed-quality information have become a top priority for knowledge demanders. As an online knowledge-sharing channel, the socialized question and answer (Q&A) platform provides important support services for knowledge payment. Based on the personal psychological dimensions of users and social capital theory, this paper aims to study the behavior mechanisms of knowledge payment users and examine the significant factors affecting user payment. Our research was conducted in two steps: a qualitative study to find these factors and a research model based on a quantitative study for testing the hypothesis. The results show that the three dimensions of individual psychology are not all positively correlated with cognitive and structural capital. Our results fill a gap in the literature on the formation of social capital in the knowledge payment environment by showing how individual psychological dimensions affect cognitive and structural capital differently. Thus, this study offers effective countermeasures for knowledge producers on social Q&A platforms to better amass their social capital. This research also makes practical recommendations for social Q&A platforms to strengthen the knowledge payment model.

## Introduction

The rise of paid Q&A platforms has supported the growth of the knowledge-sharing economy, allowing specialists from various fields to conduct online transactions using their knowledge [1–3]. A reliable survey study provided by the China Internet Network Information Center (CNNIC) found that the number of smartphone Internet subscribers in China has risen to 802 million, with 788 million of those subscribers (98.3%) connected to the Internet through mobile phones. With the rapid development of the mobile Internet, users have more diversified and convenient access to information and knowledge [4]. Knowledge has evolved into a vital tool, and because of its unique nature, it is an essential element of intellectual capital, significantly contributing to the improvement of the dynamic capabilities that can generate long-term competitiveness. These factors are critical to developing the low-carbon economy as a new potential economy [5]. People communicate and contribute knowledge through virtual communities, which are rapidly growing in popularity [6]. The knowledge-based payment

**Funding:** This research was supported by the MSIT (Ministry of Science and ICT), Korea, under the Innovative Human Resource Development for Local Intellectualization support program (IITP-2022-RS-2022-00156287) supervised by the IITP (Institute for Information & Communications Technology Planning & Evaluation). This work was supported by an IITP grant funded by the Korean government (MSIT) (No.2022-0-01203, Regional strategic Industry convergence security core talent training business). The funders had no role in the study design, data collection and analysis, publication decision, or manuscript preparation.

**Competing interests:** The authors have declared that no competing interests exist.

model has garnered interest as it can be accessed anytime and decreases the need to screen information [7]. With the explosive growth of network information, the demand for high-quality information resources has become increasingly strong [8]. Thus, methods for obtaining valuable knowledge from the vast amount of mixed-quality information have become a top priority for knowledge demanders.

As an online knowledge-sharing channel, the socialized question-and-answer platform provides essential support services for knowledge payment [9]. The major question-and-answer platforms in China have launched UGC-based knowledge payment columns, such as Zhihu Live on the Zhihu Platform and Ask Cafe on the Baidu Knowledge Platform, which have promoted the development of knowledge payment [10]. However, problems with the knowledge payment model remain, such as low-quality knowledge, serious homogenization, no content evaluation system, copyright protection, and more, which continue to influence users' desire to pay and participate. As a result, it is necessary to study user payment behavior mechanisms to promote user willingness to pay and ensure the success of the knowledge payment model.

Social capital is the sum of real and potential resources contained in a social relationship network and shared by network members. The social capital theory was first used to study the role of interpersonal relationships in community operations. It is based on the idea that interpersonal connections in online networks can create benefits. Social capital includes three dimensions: structural, relational, and cognitive [11]. The structural dimension of social capital manifests as social interaction and describes all modes of interpersonal interaction. The relational dimension of social capital pertains to identity, trust, and reciprocity and describes the resources created and expanded by the relational network formed in the interaction. The cognitive element of social capital manifests as a universal language and a shared vision, describing the resources created by members based on shared expressions and interpretations.

Many studies have shown that social capital significantly impacts the behavior of network members. The results of studies on the influence of personal motivation and social capital on the information-sharing behaviors of website users in social commerce show that social capital significantly influences the structural, relational, and cognitive dimensions [12]. Another study on the impact of social capital on users' participation enthusiasm in online group buying situations found that social capital strengthens users' interests through the intermediation of their participation enthusiasm [13]. Chang and Zhu integrated social capital with flow experience theory -an individual's overall life satisfaction can be significantly improved- to explore users' continuous application of social networking sites. Their results show that structural social capital significantly impacts the continuous use of social networking sites [14]. Chiu and other theorists studying the integration of social capital and cognition have examined the quality and quantity of knowledge sharing in virtual communities. They find that social connection, empathy, and identity impact the amount of knowledge sharing, while trust, a common language, and a common vision affect the outcomes of sharing knowledge [15]. Other studies investigating the influence of social capital and personal motivation on sharing knowledge among strangers in virtual communities have shown that altruism, identity, reciprocity, and a common language have significant positive effects on the amount of knowledge sharing [16]. Through these studies, we can clearly observe that social networks affect user behavior regarding knowledge payment, but there are few empirical studies of user knowledge payment behavior at present.

In summary, this research studies the behavior mechanisms of knowledge payment users to determine the significant factors that affect user payment for knowledge content. Therefore, in view of the shortcomings of the above research, our research questions are as follows: (1) What role does individual psychology play in creating buyer-seller social capital? (2) What is the relationship between the structural, cognitive, and relational elements of social capital? (3) How

does social capital influence knowledge payment behavior? A mixed qualitative and quantitative method was adopted to answer these questions.

## Literature review

### Social capital theory

The concept of social capital was first advanced by the American scholar Jacobs in 1961, and a more unified view was formed after its supplementation and improvement by many scholars. It is believed that "social capital is a new type of capital differing from the traditional one, which is the resource that can be acquired and utilized by being embedded in social networks to help actors achieve their goals" [17]. The measurement of social capital is central to research on this theory. At present, the most widely adopted measurement method is the three-dimensional method proposed by Nahapiet and Ghoshal, whereby social capital is measured through the structural, relational, and cognitive dimensions [11].

Among these, the structural dimension refers to the ability of individuals to establish contact with others and gain advantages from social interactions; the relational dimension is the degree of mutual trust and reciprocity among individuals; the cognitive dimension is the embodiment of common interests, values, and expressions among individuals in information exchanges. Because the social capital acquired by individuals in real or virtual worlds can reflect the characteristics of their social relations, reputations, and abilities, the theory of social capital is widely used in research on word-of-mouth and trust relationships [1, 18–24].

With the recent rapid growth of the online knowledge system, experts have paid close attention to the impact of social capital on users' shared knowledge behavior. Chung and others have studied the influencing factors of the knowledge-sharing behavior of social network users according to the characteristics of their social capital from the perspective of knowledge contributors. According to the findings, users' participation in knowledge sharing is significantly influenced by network externality, social interaction, reciprocity, and self-image display [25]. Zhao Dali and colleagues built a model of the links between social capital, attitude toward knowledge-sharing, and desire to contribute while researching knowledge-sharing behavior on WeChat Moments. It was revealed that structural, relational, and cognitive capital favorably impact users' knowledge-sharing attitudes and willingness [2, 9].

As a result, social capital is crucial in representing the social relations, reputation, and credit of network members and can aid in the development of a mutually beneficial relationship of trust among knowledge-sharing subjects. For knowledge payment, in the absence of a detailed description of knowledge products, the individual social capital reflected by real data is an important basis for paying select knowledge providers.

### Knowledge payment

The rise in the popularity of payment in exchange for knowledge brings users a new mode of knowledge sharing and communication. Although the mode of knowledge payment is becoming increasingly popular in China, research on users' knowledge payment behavior is still in its infancy. Most existing literature is limited to comparing payment modes, development trends, users' willingness to accept it, etc. According to Xu et al. (2016), while research on the influencing factors of users' knowledge payment behavior is limited [8]. The effects of performance expectancy, perceived interest, and social impact on users' knowledge-paying behavior are examined using the integrated technology acceptance model. User behavior is influenced not only by the attributes of the product itself but also by the inherent specialty and motivations of the users. Compared to the domestic knowledge payment market, overseas knowledge payment still focuses on online education, e-books, and other traditional content payments.

Payment Q&A, column subscriptions, and other emerging models are not widely used; therefore, relevant research is scarce. However, research on users' payment behavior for traditional content by foreign scholars provides us with a useful reference. Hsiao explored the willingness of social network users to pay for content based on the theory of perceived value-The capacity to satisfy a perceived need while providing satisfaction. The results suggest that users' willingness to pay is significantly affected by perceived value and switching barriers [26]. Dou (2012) discovered that the utilized value of content items substantially impacts customers' payment behavior from the perspective of perceived risk [27].

## Price

In online trading, commodity prices are one of the most important factors affecting consumers' purchasing decisions. Diecidue, Rudi, and Tang point out that according to market rules and considering cost, consumers are more inclined to purchase lower-priced commodities to avoid risks where commodity quality is difficult to discern [28]. Hustić and Gregurec (2015) further explore the impact of commodity prices on users' payment decisions and consider that the higher the price, the fewer the payers [29].

On social Q&A platforms, the price of knowledge, which is traded as a commodity, is usually set by the knowledge provider. For example, prices can range from a few yuan to hundreds of yuan for a paid Zhihu Live item on the Zhihu platform, which provides different payment level options. Therefore, based on scholars' conclusions, this paper includes knowledge payment price as a variable in the theoretical model to study its influence on users' knowledge payment behavior.

## Theoretical background and hypotheses development

In social cognitive theory, as advanced by Bandura, the triad of individuals, situations, and behaviors constitutes an interactive model that can influence behavior, and the relative influences will change in different activities and environments. This provides a framework for describing the process relationship between rational cognition and context. Among these is the objective condition that an individual acts in different social network situations. Personal cognitive factors exist in the form of situational factors, which make up cognitive, affective, and biological events [30]. These factors include perceptions of self-confidence, motivation, affective attitudes, and outcome goal orientation. Most results and actions are accompanied by these factors in interactions with the environment. This theory does not accept the view of extreme environmental determinism but rather emphasizes the influence of subjective social cognition on specific situations. In this case, social cognition refers to "how people think about themselves and the social world, in other words, how people choose, interpret, memorize and use social information to make judgments and decisions." A new outcome expectation and self-competition assessment can be formed through the absorption and study of the results and experiences so that behaviors are adjusted to develop a new behavioral result, which is a circular interactive process. According to the social cognitive theory, self-efficacy and personal result anticipation are key personal cognitive elements influencing action. "Self-efficacy" is an individual's self-confidence assessment of whether they can effectively use knowledge to perform a certain behavior. This differs from an individual's actual knowledge and ability as it is a subjective assessment from the perspective of personal cognition [31]. Numerous studies have shown that positive self-efficacy can promote the occurrence of the shared behavior of knowledge and information. Starting with self-efficacy, scholars have extended the concept of influence perception from the perspective of self-empowerment. Self-empowerment stems from social individuals' desire for internal autonomy and is the process of diminishing social

individuals' feelings of helplessness to promote self-efficacy. Self-empowerment enhances the individual initiative to take action by boosting confidence in one's abilities to create goal-achieving conditions [32]. The interactions of self-empowerment, initiative, and environment on individual behavior hinge on the discussion of the relationship of self-efficacy, personal outcome expectation, behavior, and environment in social cognitive theory. The perception of influence from the perspective of self-empowerment emphasizes the perception of capability at the psychological level rather than actual ability. The measurement of self-psychological empowerment can be divided into three levels: the ability to communicate and connect, the ability to take control of one's personal life, and the ability to influence change. Scholars have developed the concept of the perception of influence-based self-empowerment content.

An individual's psychology influences their willingness to share knowledge in virtual communities; in recent years, many studies have indicated that the self-efficacy index of knowledge sharing is the chief distinguishing feature of self-efficacy. Bock and Kim reveal that individuals' self-judgment of their contribution to an organization has a significant positive impact on knowledge sharing [33], while Kankanhalli et al in 2005 also regard self-efficacy as an internal incentive to determine its effect on knowledge sharing [34]. Previous studies have shown that individual action performance is significantly influenced by individual effect expectation, while action performance greatly affects outcome expectation [35]. Thus, we hypothesize that:

**H1**: Self-efficacy is positively associated with personal outcome expectations.

**H2**: Self-efficacy is positively associated with community-related outcome expectations.

**H3**: Self-efficacy is positively associated with shared language.

**H4**: Self-efficacy is positively associated with shared vision.

**H5**: Self-efficacy is positively associated with social interaction.

This study follows Nahapiet and Ghoshal's classical three-dimensional scale of social capital. The three dimensions of structure, relationship, and cognition measure the intensity and closeness of the relationship with self-efficacy, which is suitable for the circumstances of the social media relationship of Zhihu Live and is closer to the concept of a social relation network. From each dimension of online social capital, the study selects variables closely related to knowledge sharing as the contextual factors influencing knowledge-sharing behaviors in the research hypotheses. For example, this includes interactions in the structural dimension, trust and identity in the relational dimension, and the cognitive dimension of shared language and vision. Thus, the following hypotheses are presented:

**H6**: Personal outcome expectations are positively associated with shared language.

**H7**: Personal outcome expectations are positively associated with shared vision.

**H8**: Personal outcome expectations are positively associated with social interaction.

**H9**: Community-related outcome expectations are positively associated with t Shared language.

**H10**: Community-related outcome expectations are positively associated with shared vision.

**H11**: Community-related outcome expectations are positively associated with social interaction.

Previous research has demonstrated the link between cognitive and relational capital. Buyers and vendors who speak the same language find it easier to trust each other. Previous studies have shown that trust is likely to produce relevant behavioral norms [11, 36]. Studies have also

found that shared languages have an important impact on the terms of exchange and can help establish a mutually beneficial and respectful business relationship. As a result, cognitive capital can boost relationship capital [11, 36].

**H13**: Shared language is positively related to trust.

**H14**: Shared language is positively related to identification.

**H15**: Shared vision is positively related to trust.

**H17**: Shared vision is positively related to identification.

Structural capital is an important premise of and affects relational capital. The relationships of relational capital can be best strengthened through social interaction; thus, structural capital can enhance relational capital [37]. Trust arises from social relations and is produced from the interaction between buyers and sellers; it is the buyers that trust sellers [26, 36, 38].

**H18**: Social interaction is positively related to trust.

**H19**: Social interaction is positively related to identification.

A shared vision is a goal that community members pursue together and is also a measure of the strength that unifies the members. Users in the same subject field on the same knowledge payment platform have similar interests or personal promotion needs; that is, they hope to increase knowledge and improve their ability by paying for knowledge content. Based on these consistent values, a relationship of mutual trust is formed among users. Their shared values and common interests can establish harmonious social relations among members to promote members' identification with their social networks.

A shared language refers to the usage of jargon, acronyms, etc., by content producers and users to improve communicative efficiency. In 2011, Lu et al found that sellers use jargon to convince buyers that they will not be deceived, thereby increasing buyers' trust [36]. In knowledge payment, content producers use jargon to increase awareness of their expertise and skills, thereby building trust. In addition, the use of jargon in social networks gives users a sense of belonging and a belief that other users share their aspirations, thus generating identity.

A shared vision and language in a knowledge payment community will promote users' interactions and enhance their participation and willingness to pay. A study by Zhao and colleagues found that a sense of belonging to a virtual community can promote the intention of members to acquire knowledge [2].

**H12**: Shared language is positively related to knowledge purchase intention.

**H16**: Shared vision is positively related to knowledge purchase intention.

Topics of common interest can serve to form a relatively stable social circle, including the users and creators on knowledge payment platforms, allowing them to express their opinions on the knowledge content and engage in exchange and discussion. A social interactive connection is a channel of information and resources in social relations networks, including relationship intensity, time, frequency of interaction, etc. Lu et al have shown in 2011 that the structural dimension of social capital significantly impacts the relational dimension of social capital [36]. Frequent interactions make the relationship between members closer, thus increasing the sense of identity and belonging to the social circle. Granovetter (1985) believes that social interaction generates trust. Knowledge payment users build trust in their competence in the field of expertise through frequent interactions with content producers [39].

In addition, the emotional connection formed by the users in their interactions can promote user behavior; The closer the connection, the more users participate. Shang, Wu and Sie (2017) found that social interactive connection affects consumers' purchase intentions [40].

**H20**: Social interaction is positively related to knowledge purchase intention.

Nahapiet and Ghoshal believe that identity is a process in which individuals regard themselves as part of a whole with other individuals or groups [11], while Chiu et al. (2006) believe that identity in virtual communities is based on members' sense of belonging to the community [15]. A sense of belonging and membership will motivate users to pay. In addition, many studies have shown that trust significantly impacts consumer behavior. Hajli et al. (2017) believe that trust can encourage consumers to seek information about goods in a socialized e-commerce environment [41]. Another study found that trust significantly affects users' continuous use of mobile payment services by Zhou (2013) supporting the belief that content providers have the appropriate knowledge, experience, and skills in their field of expertise to satisfy user needs [42]. Thus, they will be willing to pay for their knowledge.

**H21**: Trust is positively related to knowledge purchase intention.

**H22**: Identification is positively related to knowledge purchase intention.

On a social Q&A platform, the knowledge price, which is traded as a commodity, is usually set by the knowledge provider. According to market rules and cost considerations, consumers are more likely to purchase goods at lower prices [28, 29]. For example, prices can range from a few yuan to hundreds of yuan for a paid item on ZhihuLive on the Zhihu platform, which provides different payment level options. Therefore, based on scholars' conclusions, this paper takes the knowledge payment price as a variable in the theoretical model to study its influence on users' knowledge payment behavior [8, 26, 27].

**H23**: The price is negatively related to knowledge purchase intention.

**H24**: The price moderates the effect of shared language on knowledge purchase intention.

**H25**: The price moderates the effect of trust on knowledge purchase intention.

**H26**: The price moderates the effect of identification on knowledge purchase intention.

**H27**: The price moderates the effect of social Interaction on knowledge purchase intention.

The conceptual research framework is constructed in (Fig 1) based on the assumed causalities between the research variables, as formulated by the above hypotheses.

## Methodology

This research was carried out in two steps. First, individual psychology and social capital factors that affect knowledge payment behavior driven by a shared economy are studied qualitatively. In this study, qualitative research was used to study the same problem using different methods so that the researchers could improve their self-confidence in the accuracy of the research results. Using qualitative research methods, a complete description of the research questions is provided to optimize the results.

Interviews are one of the most common qualitative data collection approaches, enabling researchers to gain deep insights from rich narratives [43]. Following Venkatesh and Brown (2018), to promote data reliability in a qualitative study, the interviewer asked each question in a prescribed order [44]. Meanwhile, to ensure inferential validity, including interpretive validity and confirmability, we attempted to accurately understand interviewees' thoughts, views,

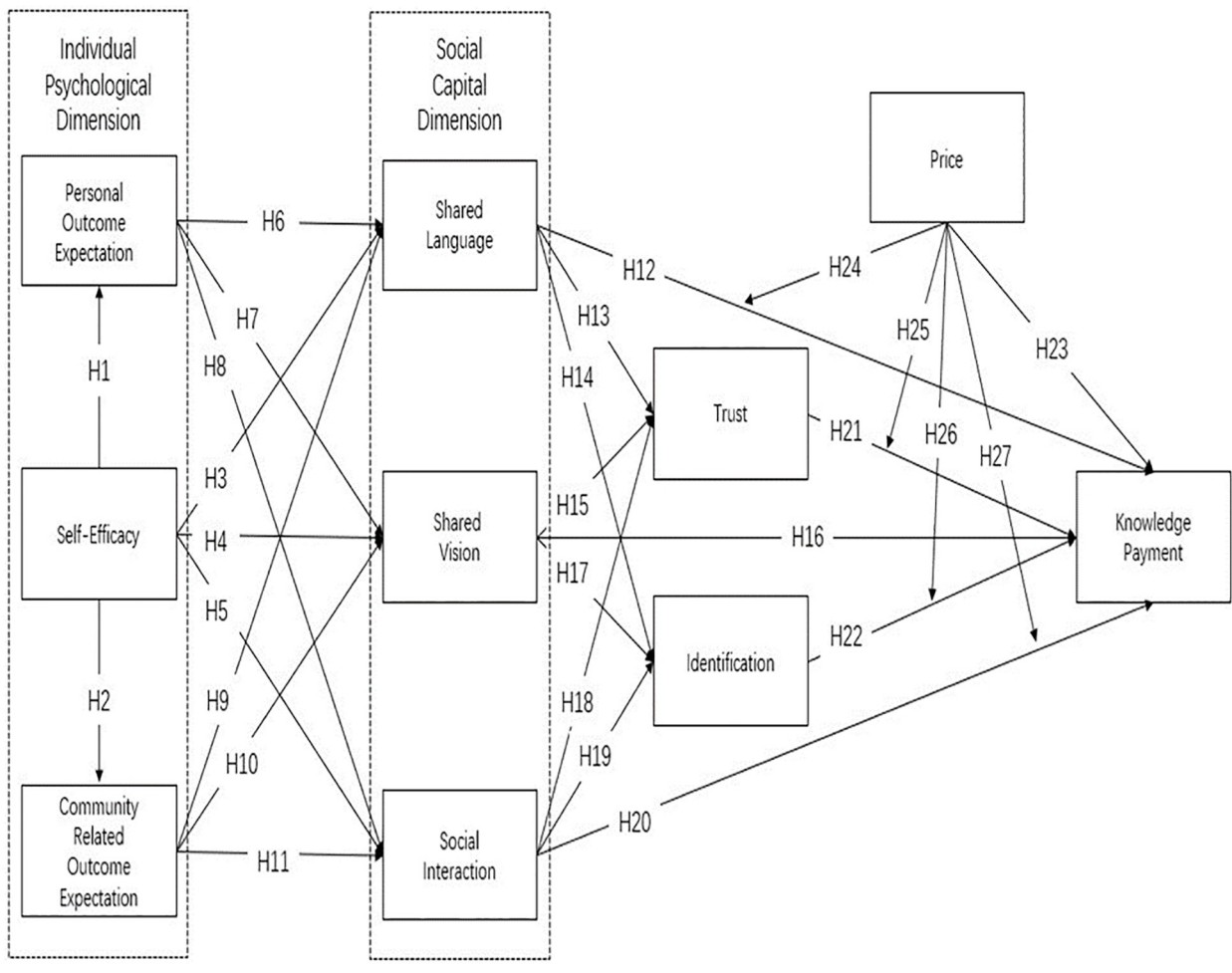

**Fig 1. The research model.**

feelings, and intentions by analyzing their discussions as recorded during the interview. We also confirmed and corroborated the interview results by cross-checking them with two colleagues to ensure their confirmability.

We used FGI to gather data on ZhihuLive users in China. We displayed posters and spread social recommendations to attract users to participate in our discussion. The focus group consisted of six users, which helped encourage a meaningful conversation [45]. The respondents consisted of four male and two female users who had 3 to 18 months of service experience, indicating that they are very familiar with the various situations in which the service is used. Their ages ranged from 20 to 30 years, representing the majority of service users. Also, the informants included university students, government and company employees, and an individual business owner.

We carefully designed the interview following Krueger and Casey (2000) [46]. The interview was carried out by a marketing lecturer and was conducted in a relaxed atmosphere in which participants were encouraged to talk openly.

The interview was divided into three sections. In the first part, we invited participants to share their service use experiences. The purpose of this initial stage was to arouse informants' enthusiasm through a light-hearted question and determine how they thought and felt about

the ZhihuLive service. In the second section, we discussed the particulars of the ZhihuLive platform and their perspectives on service through the lens of individual psychology. In the third section, we discussed the factors that influence the use of knowledge payment services, their frequency of use, and recommendations for any business model innovations. The insights from this section contribute to the subsequent development of hypotheses based on theoretical rationales from prior research.

The interview lasted nearly one hour and was smartphone-recorded and transcribed verbatim. Following the principles of Corbin and Strauss (2008) [47], the content analyses of the recording and transcript were performed separately by two colleagues unaware of the specific research framework and hypotheses to avoid potential bias.

According to the interviews and research, ZhihuLive is characterized by applications in a short time, effective utilization of this time, and an ability to provide valuable information to users to solve their problems and assist them in learning new knowledge. The users found the development of ZhihuLive promising and offered good suggestions for the application procedure, such as increasing the in-depth field knowledge, providing better growth channels for content, expanding the breadth of the content, diversifying and simplifying payments, and so forth.

ZhihuLive caters to the features of the modern Internet era, including adopting fragmentation time and knowledge payments and possessing good application performance and development prospects. However, it also has some deficiencies that should be upgraded, and it can further improve its structure and user experience during the growth process.

We invited the participants to freely state their opinions on the factors needed to enhance knowledge payment behaviors. They consistently described and confirmed the key factors of self-efficacy, personal outcome expectations, community-related outcome expectations, social interaction relations, shared languages, shared visions, trust, and identification to promote participation in knowledge payment behaviors. The interview statements are described as follows:

Self-efficacy (SE) is a dynamic construct that reflects more than just an ability assessment. An individual's judgment of SE reflects an orchestration or mobilization component that includes motivational and integrative aspects [31, 48]. As one interviewee said, "I am able to gain knowledge in ZhihuLive, and the ability to easily use it is a matter of great concern to me."

The notion that task completion leads to a certain consequence is referred to as outcome expectations. In this research, the community-related result expectation is ZhihuLive knowledge payers' judgment of the possible implications of their knowledge-paying behavior on the virtual community. The personal result expectation refers to the judgment that the ZhihuLive knowledge payer may face consequences for their knowledge payment behavior. One interviewee stated, "I had made a lot of friends in the process of learning with ZhihuLive and can study happily. My participation can help the development of the community."

Tsai investigated social interactions (network ties) as information and resource flow routes [49]. According to Granovetter (1985), relationship strength is a combination of time invested, intense emotions, closeness (mutual confidence), and reciprocal services [39]. An interviewee remarked, "In the process of learning with ZhihuLive, I receive prompt communication from the knowledge provider to address my doubts. At the end of my studies, I will take the time to listen to the communication again."

Within management literature, trust is viewed as a set of specific beliefs dealing primarily with the integrity, benevolence, and ability of another party [50]. The focus of this research is on integrity, referring to a person's expectation that individuals in a digital world will adhere to a set of widely accepted values, standards, and principles.

Chang and Chuang (2011) define identification as "one's sense of self in terms of the defining elements of a personality social category," in the case of the virtual world [16]. Identification is how individuals regard themselves as one with another person or group [11]. In this research, identification is a person's sense of belonging and favorable feelings toward the digital world, comparable to emotional identification, as suggested by Ellemers, Kortekaas and Ouwerkerk (1999) [51].

In the interview comments, one female participant said, "I paid money to sponsor this live, mainly because I wanted to get close to Big V and listen to the voice of Gejin. In the warm-up time a few days before the official start of the live event, Gejin was actively answering the questions of the live audience. It was almost a question and answer. No matter how many simple questions are answered seriously, it can be said that it is very conscientious."

Shared language addresses "the acronyms, subtleties, and underlying assumptions that are the staples of day-to-day interactions" [16]. A shared vision, according to Tsai (2022), "embodies the collective ambitions and aspirations of an organization's members" and is "A bonding technique that allows diverse areas of a company to merge or combine resources" [15, 49].

One interviewee stated, "I used to participate in a disappointing live broadcast; from the beginning to the end, it was merely a live broadcast of the host's meal in which a group of people was talking without any logic, so it could only get a negative score."

Another interviewee said, "Decoration is an area of high information asymmetry. The most valuable thing about this Live for me is that the speaker taught me the order of homework in the early stages of decoration and how to scientifically allocate time and energy."

In line with the qualitative research, these opinions demonstrate the following: first, the participants provide user engagement factors from the perspective of the psychological level, including self-efficacy, personal consequence expectation, and community-orientation consequence expectation. This interesting phenomenon offers a critical point of entry for refining the final research model.

Next, with respect to the user experience and technology, the user experience and products involve all such aspects. Apart from offering professional content, it will reinforce technological research, development, and upgrading and provide a humanized service and experience to meet more users' demands. In line with the interview results, the interviewees also expressed dissatisfaction with such matters as app design. For example, some interviewees proposed adding interactive user community elements, such as a messaging function, to facilitate discussion, which involves the horizontal and vertical structure of the app. At present, the pay-per-use content consists of simple published content or unidirectional instruction from tutors or experts, but vertical interactions between tutors and users or horizontal interactions among users are minimal, which causes a lack of community atmosphere, low attractiveness, and insufficient stickiness.

Finally, when referring to acceptable payment amounts, some users expressed that price would not be a concern if the learning content is what they need or is helpful. Others stated that with respect to affordability, they would readily accept a lower price, which is consistent with the research of Diecidue et al. (2012) [28].

Conceptual relationships related to the factors defined by qualitative research were incorporated into our research model, Fig 1. The model comprises the influencing factors of the individual psychological dimension and social capital, as well as the adjustment effect of price on knowledge payment. This was used to empirically test the relationship between these variables, as described in the next section.

The second step in our research is to test the hypothesis of the model using quantitative survey data. We focus on explaining the quantitative results based on the qualitative findings and existing frameworks.

## Measurement

A questionnaire was created to collect data on the research variables. All the multiple-item assessments in this study came from previous research and were graded on a 7-point Likert scale (1 = strongly disagree, 7 = strongly agree). The key terms of all constructs, measure items, and related sources are listed in Table 1.

The data was analyzed using SPSS 22 and Smart PLS 3.0. A three-step procedure was adopted to analyze convergent and discriminant validity and reliability. The construct scales were first fine-tuned using exploratory factor analysis (EFA) and Cronbach's reliability analysis. Factors with loading < 0.5 were excluded to ensure data quality. The link between latent components and observable items was then tested using confirmatory factor analysis (CFA). The constructs' value of average variance extracted (AVE) and composite reliability (CR) were also investigated. A preliminary assessment of the measurements is shown in Table 1.

**Table 1. Measurement items of the variables.**

| Construct Names | Measurement items (Likert 7-point scale: 1 = *Strongly disagree*; 7 = *Strongly agree*) | Adapted from |
| --- | --- | --- |
| Self-efficacy | I am confident that I can use ZhihuLive to learn. | Compeau et al. (1999); Venkateshet al. (2003) |
| | I am confident in using ZhihuLive to gain knowledge to solve problems. | |
| | I am confident that I will participate in discussions while learning using ZhihuLive. | |
| | I can use the new features provided by ZhihuLive. | |
| Personal outcome expectations | In the process of learning with Zhihu Live, I can make many friends. | Chiu et al. (2006) |
| | Learning with ZhihuLive will give me a feeling of happiness. | |
| | Using Zhihu Live to gain knowledge will give me a sense of accomplishment. | |
| | Learning with ZhihuLive will strengthen the connection between other members and me. | |
| Community-related outcome expectations | Learning with ZhihuLive will help the virtual community to operate successfully. | Chiu et al. (2006) |
| | Learning with ZhihuLive will help the community continue its operations in the future. | |
| | Learning with ZhihuLive will help the community grow. | |
| Social interaction | I have a close social relationship with content providers. | Chang & Chuang (2011) |
| | I spend a lot of time interacting with content providers. | |
| | I have frequent conversations with content providers. | |
| Trust | I believe content providers have a high level of knowledge in their areas of expertise. | Ridings, Gefen, & Arinze (2002) |
| | Content providers have relevant knowledge about their topics. | |
| | Content providers seem to be successful in their field of work. | |
| Identification | I have a sense of belonging to the community. | Chang & Chuang (2011) |
| | I feel united and intimate with the community. | |
| | I am proud to be a part of the community. | |
| Shared language | The content provider and I share a common jargon. | Chang & Chuang (2011) |
| | The content provider and I use an understandable communication model. | |
| | Content providers publish articles using understandable presentation patterns. | |
| Shared vision | Members of the community share a common learning goal. | Chiu et al. (2006) |
| | Members of the community share common values. | |
| | Members of the community share a common vision of empowering themselves by gaining knowledge in their fields of expertise. | |
| Knowledge payment | I may pay for the content in the future. | Merchant et al. (2010);Shang et al. (2017) |
| | I intend to reward content creators. | |
| | Next time, I will pay for the content. | |
| Price | Buying another course on ZhihuLive could be cheaper than this one. | Kim et al. (2009); Beneke et al. (2013) |
| | Buying another course on ZhihuLive may save more money than buying this one. | |
| | I may be spending more money on this ZhihuLive course than on another ZhihuLive course. | |

Two first-order constructs (Shared Language and Shared Vision) that are not highly associated and distinguishable were used to measure Cognitive Capital. PLS was used to estimate higher-order entities using a repeated indicator technique.

Two first-order constructs (Trust and Identification) were used to measure Relational Capital, and they were not highly associated or distinguishable. Each construct's AVE was examined.

### Data collection and analysis

The questionnaire data were divided into two parts, a preliminary evaluation and a formal investigation. A total of 86 Zhihu Live users' responses to the questionnaire were collected in April 2023 as part of the former. Participants' feedback on the questionnaire content was examined, evaluated, and revised accordingly to simplify the wording, eliminate ambiguity, and encourage descriptions of the behavior of paying for knowledge. Based on the pre-test feedback, the study tested the reliability and validity of the scale and the rationality of the questionnaire design.

The formal questionnaire survey was implemented through non-random sampling from April 5 to April 15, 2023, and was separated into offline and online channels in accordance with the requirements of actual situations. The questionnaire is available on the Questioning Star Platform. To avoid a homogeneous sample population, the author distributed the survey to WeChat users from different relationship statuses, industries, education levels, and ages through snowball diffusion on the WeChat platform. A total of 538 questionnaires were collected in this survey.

The study further screened the questionnaires to ensure their validity and credibility based on the following conditions: (1) Using the completion times of the preliminary test, the time spent completing a valid questionnaire should not be less than 180 seconds, and thus questionnaires with completion times less than this were excluded. (2) Questionnaires with inconsistent responses to similar questions were excluded. Application of the above screening criteria yielded 500 valid questionnaires, with an effective recovery rate of 92.93%.

Regarding gender and age, 58.4% of the respondents were male, with those aged 20–39 accounting for 64.4% of the total. Moreover, 69.8% of those polled had earned a college diploma or a bachelor's degree.

In terms of income, 46.2% of respondents had a monthly income of less than 5,000 CNY (1 USD = roughly 6.32 CNY), while 31.4% had a monthly income of between 5,001 and 10,000 CNY, implying that the primary target users were members of the general public.

## Results

### Sample profile

There were 292 male respondents and 208female respondents among the 500 questionnaires completed. The majority of the participants (n = 249) were between the ages of 20 and 29. Most of the students in the sample (n = 349) had attained a bachelor's degree or college diploma, followed by graduate students and students in high school or with lower education. The participants were divided into groups based on their income levels. Table 2 shows the descriptive data in greater detail.

### Measurement model

The results of the reliability and convergent validity tests are shown in Table 3. The CR and Cronbach's values for all constructs were greater than 0.8, showing strong scale reliability and

**Table 2. Demographics of respondents (*n* = 500).**

| Category | Item | Frequency | Percentage |
|---|---|---|---|
| Gender | Male | 292 | 58.4 |
| | Female | 208 | 41.6 |
| Age | < 20 | 100 | 20 |
| | 20–29 | 249 | 49.8 |
| | 30–39 | 93 | 18.6 |
| | 40–49 | 41 | 8.2 |
| | >50 | 17 | 3.4 |
| Education | High school or lower | 37 | 7.4 |
| | Bachelor's degree or college | 349 | 69.8 |
| | Graduate degree | 114 | 22.8 |
| Income (Monthly, CNY) | < 5,000 | 231 | 46.2 |
| | 5,001–10,000 | 157 | 31.4 |
| | 10,001–15,000 | 78 | 15.6 |
| | >15,000 | 34 | 6.8 |

validity [52, 53]. In terms of convergent validity, the standardized factor loadings of indicators were significantly greater than 0.7 for all constructs. The values of CR were higher than 0.7, and the values of average variance extracted (AVE) for all constructs exceeded the recommended minimum of 0.5, indicating satisfactory convergent validity [52, 53].

We compared the square root of the AVE for each construct with the inter-construct correlation values for all construct pairings to verify discriminant validity, as indicated by (45). Table 4 shows the construct correlation estimates and the square roots of AVE (bold diagonal elements) for the constructs. The square roots of AVE are all larger than the other items in each row and column, indicating that discriminant validity is sufficient.

We also looked at the variance inflation factor (VIF) values for antecedent variables to see whether there was any potential for multicollinearity, as recommended by Tabachnick and Fidell in 1996 [54]. The VIF values, which ranged from 1.1 to 2.8, did not exceed the threshold value of 10.0, indicating that multicollinearity was not a significant issue in this study.

In addition, a common method bias (CMB) may exist in self-reported data from a single source, jeopardizing the study's validity. We employed the unmeasured latent method construct (ULMC) methodology in PLS to estimate the CMB amount, following Liang et al. (2007) [55]. Table 5 displays the common method bias test results, which indicate that the average substantively explained variance of indicators is 0.826, whereas the average method-based variance of the indicators is 0.001. The ratio of substantive variance to method variance is quite large (i.e., 826:1). Meanwhile, most method factor loadings are insignificant (only one is significant). As indicated by the low volume and insignificance of technique variance, CMB was not a critical concern in our investigation.

## Hypotheses test results

We utilized Smart PLS 3.0 to do a path analysis to test the study hypotheses that is shown in (Fig 2). First, among the factors of individual psychological dimension, Self-Efficacy was found positively affect Personal Outcome Expectations and social interaction (β = 0.107, p < 0.05 and β = 0.134, p < 0.001, respectively), supporting H1 and H5. However, it had no effect on Community-Related Outcome Expectations and Shared language and Shared vision (β = 0.032, p > 0.05 and β = -0.029, β = 0.053, p > 0.05, respectively); Thus, H2 and H3 and H4 were not supported. Personal Outcome Expectations was found to positively affect Shared

**Table 3. Results of the reliability and convergent validity tests.**

| Construct | Indicator | Standardized loading | Cronbach's α | CR | AVE |
|---|---|---|---|---|---|
| Self-efficacy | SE1 | 0.909 | 0.930 | 0.940 | 0.825 |
| | SE2 | 0.897 | | | |
| | SE3 | 0.895 | | | |
| | SE4 | 0.931 | | | |
| Personal outcome expectations | POE1 | 0.910 | 0.921 | 0.924 | 0.808 |
| | POE2 | 0.897 | | | |
| | POE3 | 0.895 | | | |
| | POE4 | 0.894 | | | |
| Community-related outcome expectations | COE1 | 0.887 | 0.863 | 0.864 | 0.786 |
| | COE2 | 0.869 | | | |
| | COE3 | 0.903 | | | |
| Shared language | SL1 | 0.925 | 0.903 | 0.905 | 0.838 |
| | SL2 | 0.900 | | | |
| | SL3 | 0.921 | | | |
| Shared vision | SV1 | 0.930 | 0.907 | 0.908 | 0.843 |
| | SV2 | 0.905 | | | |
| | SV3 | 0.920 | | | |
| Trust | TR1 | 0.888 | 0.888 | 0.889 | 0.817 |
| | TR2 | 0.911 | | | |
| | TR3 | 0.912 | | | |
| Identification | ID1 | 0.872 | 0.864 | 0.865 | 0.786 |
| | ID2 | 0.880 | | | |
| | ID3 | 0.907 | | | |
| Social interaction | SI1 | 0.889 | 0.876 | 0.881 | 0.802 |
| | SI2 | 0.893 | | | |
| | SI3 | 0.904 | | | |
| Knowledge payment | KP1 | 0.893 | 0.852 | 0.852 | 0.771 |
| | KP2 | 0.869 | | | |
| | KP3 | 0.873 | | | |
| Price | PR1 | 0.936 | 0.918 | 0.919 | 0.859 |
| | PR2 | 0.913 | | | |
| | PR3 | 0.932 | | | |

Language ($\beta = 0.549$, $p < 0.001$), Shared vision and Social Interaction ($\beta = 0.064$, $p<0.05$ and $\beta = 0.089$, $p<0.05$), suggesting that H6, H7 and H8 were supported. Meanwhile, Community-Related Outcome Expectations was found to have a positive effect on Shared Language and Shared vision ($\beta = 0.256$, $p<0.001$ and $\beta = 0.532$, $p<0.001$) but no effect on Social Interaction ($\beta = 0.061$, $p>0.05$), indicating that H9 and H10 was supported but H11 was not.

Second, while Shared Language had favorably influenced Knowledge Payment, Trust and Identification ($\beta = 0.273$, $p<0.001$, $\beta = 0.092$, $p<0.05$ and $\beta = 0.255$, $p<0.001$, respectively), implying that H12, H13 and H14 were supported.

Moreover, Shared vision positively affected on Trust, Knowledge Payment and Identification ($\beta = 0.200$, $p<0.001$, $\beta = -0.188$, $p<0.001$ and $\beta = 0.274$, $p<0.001$, respectively), supporting H15, H16 and H17. Social Interaction favorably influenced on Trust, Identification and Knowledge Payment ($\beta = 0.157$, $p<0.001$ and $\beta = 0.230$, $p<0.001$ and $\beta = 0.278$, $p<0.001$), supporting H18 and H19and H20. Trust positively affected Knowledge Payment ($\beta = 0.204$,

**Table 4. Construct correlations and discriminant validity.**

|       | COE   | ID    | KP    | POE   | PR    | SE    | SI    | SL    | SV    | TR    |
|-------|-------|-------|-------|-------|-------|-------|-------|-------|-------|-------|
| COE   | 0.887 |       |       |       |       |       |       |       |       |       |
| ID    | 0.143 | 0.887 |       |       |       |       |       |       |       |       |
| KP    | 0.031 | 0.489 | 0.878 |       |       |       |       |       |       |       |
| POE   | 0.015 | 0.254 | 0.139 | 0.899 |       |       |       |       |       |       |
| PR    | 0.046 | 0.420 | 0.166 | 0.091 | 0.927 |       |       |       |       |       |
| SE    | 0.043 | 0.276 | 0.248 | 0.115 | 0.259 | 0.908 |       |       |       |       |
| SI    | 0.074 | 0.320 | 0.402 | 0.112 | 0.533 | 0.157 | 0.896 |       |       |       |
| SL    | 0.286 | 0.360 | 0.225 | 0.595 | 0.024 | 0.040 | 0.086 | 0.915 |       |       |
| SV    | 0.601 | 0.385 | 0.221 | 0.072 | 0.020 | 0.085 | 0.120 | 0.184 | 0.918 |       |
| TR    | 0.063 | 0.341 | 0.540 | 0.162 | 0.098 | 0.200 | 0.207 | 0.154 | 0.258 | 0.904 |

COE: Community-related outcome expectations, ID: Identification, KP: Knowledge payment, POE: Personal outcome expectations, PR: Price, SE: Self-efficacy, SI: Social interaction, SL: Shared language, SV: Shared vision, TR: Trust

p<0.001), supporting H21. Identification insignificantly influenced Knowledge Payment (β = 0.045, p>0.05), not supporting H22. Price was not significantly related to Knowledge Payment (β = 0.018, p>0.05), not supporting H23. Price had moderating effect of Shared Language, Trust and Identification on Knowledge Payment (β = -0.522, p<0.001, β = 0.195, p<0.001, β = -0.086, p<0.05), supporting H24, H25, H26. Price had no moderating effect on Social Interaction and Knowledge Payment (β = 0.086, p>0.05), showing that H27 was not supported. "Table 6" shows the summary table reporting the acceptance or rejection of each hypothesis:

## Discussion and conclusions

Within the framework of social cognition theory, in combination with the concept of social capital, this study explores the mechanism of the influence of individual psychological dimensions and social capital on users' knowledge payment behavior based on specialized theories. At the individual psychological level, the study mainly selects self-efficacy, individual outcome expectation, and community outcome expectation as influential factors. The results of the research show that self-efficacy positively impacts the structural dimensions of personal outcome expectations and social capital, while personal outcome expectations and community outcome expectations positively impact the cognitive dimensions of social capital. At the same time, the cognitive dimensions also positively affect the relational dimension. The structural dimension positively impacts the relational dimension and knowledge payment, and the relational dimension positively affects knowledge payment.

In this paper, the empirical results show a significant difference between self-efficacy & personal expectations and social capital structure. That is, in a Q&A community, the more confident a user is in their ability to complete a certain task or engage in a behavior, the more frequent communication they will have with others in the community, which will positively stimulate the individual's expected results.

In the Q&A community, knowledge sharers should judge their abilities based on comprehensive evaluation criteria, including their ability to use the Q&A community and to express their opinions and expertise. The above analysis shows that self-efficacy is a form of self-assessment of the users' skills and confidence in completing particular behaviors. If the users are recognized for their shared information, they will be more confident and continue to share more knowledge.

**Table 5. Results of the common method bias test.**

| Construct | Indicator | Standardized Factor loading(R) | $R^2$ |
|---|---|---|---|
| **Self-efficacy** | SE1 | 0.909*** | 0.826 |
| | SE2 | 0.897*** | 0.805 |
| | SE3 | 0.895*** | 0.801 |
| | SE4 | 0.931*** | 0.867 |
| **Personal outcome expectations** | POE1 | 0.910*** | 0.828 |
| | POE2 | 0.897*** | 0.805 |
| | POE3 | 0.895*** | 0.801 |
| | POE4 | 0.894*** | 0.799 |
| **Community-related outcome expectations** | COE1 | 0.887*** | 0.787 |
| | COE2 | 0.869*** | 0.755 |
| | COE3 | 0.903*** | 0.815 |
| **Shared language** | SL1 | 0.925*** | 0.856 |
| | SL2 | 0.900*** | 0.810 |
| | SL3 | 0.921*** | 0.848 |
| **Shared vision** | SV1 | 0.930*** | 0.865 |
| | SV2 | 0.905*** | 0.819 |
| | SV3 | 0.920*** | 0.846 |
| **Trust** | TR1 | 0.888*** | 0.789 |
| | TR2 | 0.911*** | 0.830 |
| | TR3 | 0.912*** | 0.832 |
| **Identification** | ID1 | 0.872*** | 0.760 |
| | ID2 | 0.880*** | 0.774 |
| | ID3 | 0.907*** | 0.823 |
| **Social interaction** | SI1 | 0.889*** | 0.790 |
| | SI2 | 0.893*** | 0.797 |
| | SI3 | 0.904*** | 0.817 |
| **Knowledge payment** | KP1 | 0.893*** | 0.797 |
| | KP2 | 0.869*** | 0.755 |
| | KP3 | 0.873*** | 0.762 |
| **Price** | PR1 | 0.936*** | 0.876 |
| | PR2 | 0.913*** | 0.834 |
| | PR3 | 0.932*** | 0.869 |
| **Average** | | | 0.826 |

*$p < 0.05$,

**$p < 0.01$,

***$p < 0.001$

The empirical analysis shows that community users' self-efficacy significantly influences outcome expectations. Usually, those with higher self-efficacy will have more expectations of knowledge sharing. Research by Shang Yonghui, Ai Zhongqi, and Wang Fengyan on knowledge-sharing behaviors in virtual communities shows a significantly positive correlation between self-efficacy and outcome expectation.

In this paper, the empirical results show that outcome expectation significantly influences the cognitive level of social capital but does not significantly influence the.

The correlational research on the factors influencing knowledge sharing demonstrates that outcome expectation has no significant effect at the structural level. The two reasons for this

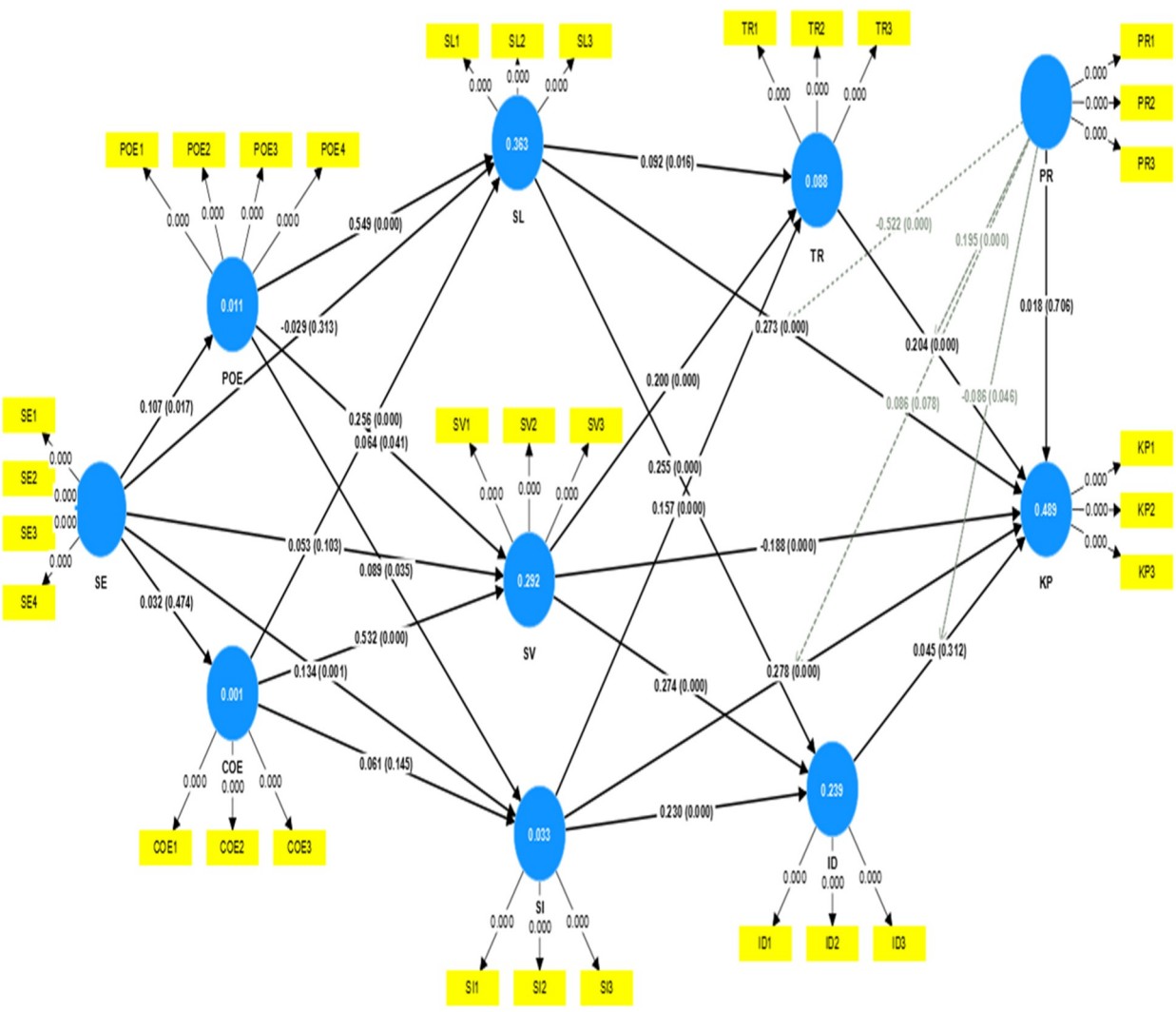

**Fig 2. Hypotheses test results.**

are suggested to be that: (1) Q&A community members are weakly linked. As there is no face-to-face communication among members, it is difficult to establish good relationships, and it is easy for connections to be destroyed. The expected return on relationships, such as praise, support, respect, trust, internal satisfaction, or a sense of accomplishment from helping others, cannot be guaranteed; (2) within the community, users share knowledge without utilitarian purposes. Most knowledge-sharing behavior is done solely based on the value or characteristics of the knowledge itself.

In a knowledge-paying community based on common needs or interests, users share the same learning goals and values and form a consistent view of problems; thus, they generate a sense of identity through high-quality communications. The jargon and professional abbreviations used by content producers and users significantly affect building trust among users. In the knowledge-sharing behavior of virtual communities, the jargon and professional abbreviations can help users develop a more effective understanding of information.

Since specific professional sectors have their own unique professional terms, the application of these terms enables users to believe that the content is professional and thus builds their

**Table 6. Summary table reporting the acceptance or rejection of each hypothesis.**

| Hypothesis | Path | Beta Score | Result |
|---|---|---|---|
| H1 | SE->POE | β = 0.107, p<0.05 | Accepted |
| H2 | SE->COE | β = 0.032, p>0.05 | Rejected |
| H3 | SE->SL | β = -0.029, p>0.05 | Rejected |
| H4 | SE->SV | β = 0.053, p>0.05 | Rejected |
| H5 | SE->SI | β = 0.134, p<0.001 | Accepted |
| H6 | POE->SL | β = 0.549, p<0.001 | Accepted |
| H7 | POE->SV | β = 0.064, p<0.05 | Accepted |
| H8 | POE->SI | β = 0.089, p<0.05 | Accepted |
| H9 | COE->SL | β = 0.256, p<0.001 | Accepted |
| H10 | COE->SV | β = 0.532, p<0.001 | Accepted |
| H11 | COE->SI | β = 0.061, p>0.05 | Rejected |
| H12 | SL->KP | β = 0.273, p<0.001 | Accepted |
| H13 | SL->TR | β = 0.092, p<0.05 | Accepted |
| H14 | SL->ID | β = 0.255, p<0.001 | Accepted |
| H15 | SV->TR | β = 0.200, p<0.001 | Accepted |
| H16 | SV->KP | β = -0.188, p<0.001 | Accepted |
| H17 | SV->ID | β = 0.274, p<0.001 | Accepted |
| H18 | SI->TR | β = 0.157, p<0.001 | Accepted |
| H19 | SI->ID | β = 0.230, p<0.001 | Accepted |
| H20 | SI->KP | β = 0.278, p<0.001 | Accepted |
| H21 | TR->KP | β = 0.204, p<0.001 | Accepted |
| H22 | ID->KP | β = 0.045, p>0.05 | Rejected |
| H23 | PR->KP | β = 0.018, p>0.05 | Rejected |
| H24 | PR x SL->KP | β = -0.522, p<0.001 | Accepted |
| H25 | PR x TR->KP | β = 0.195, p<0.001 | Accepted |
| H26 | PR x ID->KP | β = -0.086, p<0.05 | Accepted |
| H27 | PR x SI->KP | β = 0.086, p>0.05 | Rejected |

trust in the expertise of the content producer. On Zhihu, the knowledge community based on UGC (user-generated content), there are many professionals from different sectors. Since they generate professional content and common language used in discussions, professionals have a good reputation among users, leading to the users' willingness to use Zhihu Live knowledge acquisition.

Shared language and vision have no significant effect on users' intention to pay. The possible reasons for this include: (1) The common language used by knowledge producers while generating content affects the users' perception of their professional competence, integrity, and goodwill. That is, trust will affect users' intent to pay for knowledge. (2) As an emerging sector, existing users pay for knowledge to solve problems and improve their capabilities. Therefore, among the factors affecting payment behavior, the users' judgment of the expertise of the knowledge producer is greater than the relationship between the users.

## Theoretical implications

This study offers several theoretical implications. First, this research reveals important antecedents for users to participate on a social network platform. From the perspective of individual psychology and social capital theory, this involves the social capital framework against a

background of common and newly developed knowledge [17, 56], which makes it one of the few studies to apply Nahapiet and Goshal's sociological analysis, as far as we know [11].

In the theory of social cognition, the relationship status of interpersonal communication as a situational factor will also have a promotional effect on knowledge-sharing behavior. However, the specific definition and explanation of the component elements of social networks have not been given in social cognitive theory. As the meaning and operation of social capital in the field of sociology are like the concept of a social relations network, the theory of social capital can be incorporated into the social cognitive theory. The attributes of social communication in the context of the Zhihu Live platform highlight the existence of social capital. Social capital can promote the participation of Zhihu Live users on social media because part of their social capital is generated and maintained during interactions between individuals and others on social issues.

Second, previous studies mainly focused on online community knowledge sharing, community participation, and social network use [56–59]. However, there has been no research on establishing social capital regarding the public ownership of knowledge. This prevents us from understanding the mechanism of the formation and development of social capital in sufficient depth. Our research results fill this gap in the literature by emphasizing the importance of individual psychological factors that affect the formation of social capital.

Third, considering the significant negative impact of price on users' knowledge payment behavior, the Q&A platform should focus more on the professional background, knowledge level, and credibility of knowledge providers with higher price settings. This will help users make rational judgments and ensure that more professional knowledge providers are actively recommended to users with higher-paying capacities.

## Practical implications

The research results yield the following advice for knowledge-paying platforms: (1) Platforms should provide technical support and optimize interface design to promote communication between knowledge producers and users to help build a strong social relationship, increasing users' willingness to pay. (2) It is necessary to establish a content recommendation and screening system to determine the areas of user interest more accurately and help users make good matches to discover groups of like-minded people. A knowledge payment community with a shared vision is then likely to form among the knowledge content producers and ordinary users. (3) It is essential to establish user appraisal and after-sale appeal mechanisms, which can encourage knowledge producers to create high-quality content while simultaneously regulating their behavior, reducing the risk of negative user perceptions and enhancing trust.

Social Q&A platforms encounter many suppliers who wish to gain surplus knowledge through knowledge payment and obtain high-quality content. On the one hand, the social network can establish and enhance social relationships between the supply and demand sides. Highly competent knowledge providers who actively participate can be supported in accumulating more social capital to expand their popularity and influence. This would also allow knowledge demanders to broaden the scope of payment object choices through a wide range of social connections. On the other hand, the Q&A platform should establish a payment audit and appraisal mechanism to ensure quality by scientifically evaluating the qualifications and professional abilities of the knowledge providers to increase the willingness of users to engage in knowledge payment. In addition, a measurement index reflecting the knowledge provider's social capital can be included to help accurately recommend domain experts to users. At the same time, considering the significant negative impact of price on users' knowledge payment

behavior, the Q&A platform should focus more on knowledge providers' professional backgrounds, knowledge levels, and credibility. This will help users make rational judgments and ensure that more professional knowledge providers are actively recommended to users with higher-paying capacities.

## Limitations and future research

This paper provides theoretical support and corresponding business strategies for the development and operation of platforms by studying the factors affecting the willingness of online platform users. However, due to the limited scientific research capabilities and conditions, the study of related factors may be insufficient.

1. Due to the constraints of time and space, this study used online questionnaires. Although relevant filling requirements were set to ensure the effectiveness of the questionnaire, virtuality and network uncertainty weakened the quality of the questionnaire, and the responder groups were somewhat limited. Therefore, future research should conduct questionnaires using more offline channels to ensure the objectivity and validity of the questionnaire data as much as possible.

2. Although this study's structural equation model is based on the results of previous scholars' research, and the data collected has been confirmed to be reliable and valid, the selection of model variables was made solely based on individual learning and research experience. Thus, the study has not yet determined all the variables that affect users' willingness to pay. Therefore, in future research and studies, research models on the influencing factors of users' willingness to pay should be further improved.

3. The research on online knowledge service platforms is conducted from a macro perspective. In reality, online platforms have different operation methods. Therefore, this study lacks research on platform personality. Future relevant theoretical research can study a specific platform for more targeted exploration.

In future research, we will consider cooperation with Baidu Knows and other social Q&A platforms to obtain more effective data for measuring social capital so that the dimensions of research on user knowledge payment behavior can explore more key factors affecting this behavior. In addition to social capital, other factors, such as perceived value and user experience, will also affect users' payment intentions for knowledge, which must be considered in future studies. At the same time, the empirical research object must be expanded to improve the applicability and generalizability of the theoretical model.

## Supporting information

**S1 Data.**
(ZIP)

## Acknowledgments

First and foremost, we extend our heartfelt thanks to our research collaborators and co-authors for their invaluable insights, dedication, and support equally throughout the entire process. Their expertise and contributions have greatly enriched this study.

## Author Contributions

**Conceptualization:** Qin Ying.

**Data curation:** Qin Ying, Md. Mukitul Hoque.

**Formal analysis:** Qin Ying, Md. Mukitul Hoque.

**Funding acquisition:** Sang-Joon Lee.

**Investigation:** Qin Ying, Md. Mukitul Hoque.

**Methodology:** Sang-Joon Lee.

**Project administration:** Sang-Joon Lee.

**Software:** Qin Ying, Md. Mukitul Hoque, Sang-Joon Lee.

**Supervision:** Sang-Joon Lee.

**Validation:** Qin Ying.

**Visualization:** Md. Mukitul Hoque.

**Writing – original draft:** Qin Ying, Md. Mukitul Hoque.

**Writing – review & editing:** Md. Mukitul Hoque, Sang-Joon Lee.

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
