## [Decision Letter · Decision Letter 0]

27 Sep 2022

PONE-D-22-09610What Factors Determine Users' Knowledge Payment Decisions? A Mixed-Method StudyPLOS ONE

Dear Dr. Lee,

Thank you for submitting your manuscript to PLOS ONE. After careful consideration, we feel that it has merit but does not fully meet PLOS ONE’s publication criteria as it currently stands. Therefore, we invite you to submit a revised version of the manuscript that addresses the points raised during the review process.

We look forward to receiving your revised manuscript.

Kind regards,

Luigi Cembalo, PhD

Academic Editor

PLOS ONE

2.You indicated that ethical approval was not necessary for your study. We understand that the framework for ethical oversight requirements for studies of this type may differ depending on the setting and we would appreciate some further clarification regarding your research. Could you please provide further details on why your study is exempt from the need for approval and confirmation from your institutional review board or research ethics committee (e.g., in the form of a letter or email correspondence) that ethics review was not necessary for this study? Please include a copy of the correspondence as an ""Other"" file.

4.Thank you for stating the following in the Acknowledgments Section of your manuscript:

“This research was made possible by a funding from the Korean government (MSIT)'s-(No. 2019-0-01343)- Institute of Information and Communications Technology Planning and Evaluation (IITP).”

“This research was funded by supported by the BK21 FOUR Program, funded by the

Ministry of Education (MOE, Korea) and the National Research Foundation of Korea (NRF).”

5.Thank you for stating the following financial disclosure:

“This research was funded by supported by the BK21 FOUR Program, funded by the

Ministry of Education (MOE, Korea) and the National Research Foundation of Korea (NRF).”

6.In your Data Availability statement, you have not specified where the minimal data set underlying the results described in your manuscript can be found. PLOS defines a study's minimal data set as the underlying data used to reach the conclusions drawn in the manuscript and any additional data required to replicate the reported study findings in their entirety. All PLOS journals require that the minimal data set be made fully available. For more information about our data policy, please see http://journals.plos.org/plosone/s/data-availability.

7.PLOS requires an ORCID iD for the corresponding author in Editorial Manager on papers submitted after December 6th, 2016. Please ensure that you have an ORCID iD and that it is validated in Editorial Manager. To do this, go to ‘Update my Information’ (in the upper left-hand corner of the main menu), and click on the Fetch/Validate link next to the ORCID field. This will take you to the ORCID site and allow you to create a new iD or authenticate a pre-existing iD in Editorial Manager. Please see the following video for instructions on linking an ORCID iD to your Editorial Manager account: https://www.youtube.com/watch?v=_xcclfuvtxQ.

Reviewers' comments:

Reviewer's Responses to Questions

**Comments to the Author**

1. Is the manuscript technically sound, and do the data support the conclusions?

Reviewer #1: Yes

Reviewer #2: Yes

2. Has the statistical analysis been performed appropriately and rigorously? 

Reviewer #1: Yes

Reviewer #2: Yes

3. Have the authors made all data underlying the findings in their manuscript fully available?

Reviewer #1: Yes

Reviewer #2: No

4. Is the manuscript presented in an intelligible fashion and written in standard English?

Reviewer #1: No

Reviewer #2: No

5. Review Comments to the Author

Reviewer #1: Thank you for the opportunity to review this manuscript. The paper entitled "What Factors Determine Users' Knowledge Payment Decisions? A Mixed-Method Study" examines which psychological dimensions affect users' willingness to pay for knowledge content in the context of paid Q&A platforms. Overall, the paper is interesting and proposes an innovative and complex model to understand knowledge payment decisions. However, I suggest making some revisions to make the manuscript clearer and more precise.

Below I list some comments/suggestions that the authors may consider to further strengthen their manuscript.

- The article would benefit from a further revision by an English speaker and a copyeditor. Moreover, there are some typing errors, such as the absence of spaces between words throughout the entire manuscript.

- About the study design, the authors declare that they have conducted a mixed-method study, also specifying it in the title. However, the results of the qualitative study are not even hinted at. Based on the title of the paper, one would expect to read both qualitative and quantitative results. For this reason, I suggest changing the title of the paper or at least synthesizing what emerged from the qualitative study. In fact, all the variables that the authors have chosen to include in the model seem strongly anchored to the literature on the subject; therefore, one would wonder what contribution the qualitative study gave to the elaboration of the tested model.

- "Introduction" and "Literature review" are clear, however the authors might think of giving a more precise definition of the theories they cite from time to time, rather than taking them for granted, for example "flow experience theory" (p. 3, line 91) and "theory of perceived value" (p. 5, line 179).

- P. 6, lines 182-193, the authors state "Dou (2014) discovered that the utilised value of content items had a substantial impact on customers' payment behavior from the perspective of perceived risk". What do the authors mean by "perspective of perceived risk"?

- P. 7, lines 245-249, the authors state "An individual's psychology has an influence on his or her willingness to share knowledge in the virtual community; in recent years, many studies have indicated that the self-efficacy index of knowledge sharing is the chief distinguishing feature of self efficacy. Bock and Kim reveal that individuals' self-judgment of their contribution to an organization has a significant positive impact on knowledge sharing (Bock & Kim, 2002), while Atreyi,Tanbcy, and Wei also regard self-efficacy as an internal incentive to determine its effect on knowledge sharing (Atreyi et al., 2005)." It is not clear why the authors focus on the self-efficacy of knowledge providers and their knowledge sharing behavior if the focus of the research is on the sense of self-efficacy experienced by users (i.e., being confident of gaining relevant knowledge from the platform).

- P. 12, line 479, I think the authors should correct "variable inflation factor (VIF)" into "variance inflation factor (VIF)".

- P. 13, "Hypotheses test results" paragraph. Did the authors control for the effect of demographic characteristics on the estimated paths in testing the structural relationships?

- P. 14, "Discussion and Conclusions" paragraph. Before discussing the theoretical and practical implications of the study, the results of the analyzes (including non-significant ones) should be discussed and justified. Instead, the authors just summarize them. The reader may be interested in why some relationships didn't work out, for example, the reasons why self-efficacy does not affect community-related outcome expectations and cognitive dimension. Therefore, I suggest arguing all the results obtained.

- P. 16, "Limitations and future research" paragraph. Authors should recall the study's limitations, such as the use of self-report measures, the focus on intent to pay rather than actual user behavior, and the specificity of the target investigated (most participants were young adults and students), which could limit the generalizability of the results obtained.

Reviewer #2: The manuscript “What Factors Determine Users' Knowledge Payment Decisions? A Mixed-Method Study" explores the dimensions affecting willingness to pay for knowledge content in Q&A platforms. The research questions are surely relevant, and the development of the study effectively tackles core issues, for scholars and practitioners. I believe the paper is worth publication, after few major and minor amendments. Indeed, most of my suggestions are only devoted to enriching the overall readability and clarity of the paper.

Major remarks

It is not clear why Authors refer to mixed-method study if the entire manuscript presents only the results of a quantitative analysis. A couple of times it is cited a qualitative study, which outcomes however are never depicted. I invite Authors to provide full information about the qualitative study (and its contribution to the research development) or to modify the title (and narrative of the manuscript), clarifying the utility (and inputs) of the qualitative study.

Clarify the sample characteristics. It is currently reported as “non-random”, does it mean it is a convenience sample? Please state so. Additionally, in the results section (sample profile) it is stated [lines 456-457]: “The majority of the students in the sample (n = 349) studied for a Bachelor's degree or college..” Which leads readers to think that the sample is entirely formed by students (partially supported by the overwhelming majority of respondents aged between 20 and 39). Is this the case? Furthermore, the shortcomings stemming from a non-representative sample should be clearly reported in the results, discussion and limitations sections.

Minor remarks

I suggest that the manuscript receives professional English editing (several sentences are quite unclear, and the development is not easy to grasp).

Having seventeen research hypotheses, I would strongly advice Authors to add a summary table reporting the acceptance or rejection of each hypothesis.

Authors should provide a through description of the core limitations of the research (in the current version only future research are foreseen).

No information on ethical approval (or waiver) of the study is currently provided in the manuscript, neither information on the collection of informed consent from respondents (or any other ethical guideline followed by the researchers).

6. PLOS authors have the option to publish the peer review history of their article (what does this mean?). If published, this will include your full peer review and any attached files.

Reviewer #1: No

Reviewer #2: No

---

## [Author Response · Author response to Decision Letter 0]

27 Apr 2023

Dear Reviewers,

Thank you for allowing us to revise the manuscript and address the reviewers’ comments.

Please see the response letter, including our point-by-point responses to the editor and reviewers’ comments.

We hope that you are satisfied with this letter and that the manuscript will now be suitable for publication. 

Sincerely, 

On behalf of all authors, 

Prof. Dr. Sang-Joon Lee

---

## [Decision Letter · Decision Letter 1]

8 Jun 2023

What Factors Determine Users' Knowledge Payment Decisions? A Mixed-Method Study

PONE-D-22-09610R1

Dear Dr. Lee,

We’re pleased to inform you that your manuscript has been judged scientifically suitable for publication and will be formally accepted for publication once it meets all outstanding technical requirements.

Kind regards,

Luigi Cembalo, PhD

Academic Editor

PLOS ONE

Reviewers' comments:

Reviewer's Responses to Questions

**Comments to the Author**

1. If the authors have adequately addressed your comments raised in a previous round of review and you feel that this manuscript is now acceptable for publication, you may indicate that here to bypass the “Comments to the Author” section, enter your conflict of interest statement in the “Confidential to Editor” section, and submit your "Accept" recommendation.

Reviewer #1: All comments have been addressed

Reviewer #2: All comments have been addressed

2. Is the manuscript technically sound, and do the data support the conclusions?

Reviewer #1: Yes

Reviewer #2: Yes

3. Has the statistical analysis been performed appropriately and rigorously? 

Reviewer #1: Yes

Reviewer #2: Yes

4. Have the authors made all data underlying the findings in their manuscript fully available?

Reviewer #1: Yes

Reviewer #2: Yes

5. Is the manuscript presented in an intelligible fashion and written in standard English?

Reviewer #1: Yes

Reviewer #2: Yes

6. Review Comments to the Author

Reviewer #1: The authors responded exhaustively to all the questions raised and revised the manuscript accordingly. Thank you.

Reviewer #2: Authors have addressed all the raised comments provided in Round 1 in a convincing manner. Well done

7. PLOS authors have the option to publish the peer review history of their article (what does this mean?). If published, this will include your full peer review and any attached files.

Reviewer #1: No

Reviewer #2: No

---

## [Editor Report · Acceptance letter]

26 Jun 2023

PONE-D-22-09610R1 

What factors determine users’ knowledge payment decisions? A mixed-method study 

Dear Dr. Lee:

I'm pleased to inform you that your manuscript has been deemed suitable for publication in PLOS ONE. Congratulations! Your manuscript is now with our production department. 

Kind regards, 

on behalf of

Dr. Luigi Cembalo 

Academic Editor

PLOS ONE